# Magnetostriction Enhancement in Midrange Modulus Magnetorheological Elastomers for Sensor Applications

**DOI:** 10.3390/mi14040767

**Published:** 2023-03-29

**Authors:** Muhammad Asyraf Tasin, Siti Aishah Abdul Aziz, Saiful Amri Mazlan, Mohd Aidy Faizal Johari, Nur Azmah Nordin, Shahir Yasin Mohd Yusuf, Seung-Bok Choi, Irfan Bahiuddin

**Affiliations:** 1Engineering Materials and Structures (eMast) iKohza, Malaysia-Japan International Institute of Technology (MJIIT), Universiti Teknologi Malaysia, Kuala Lumpur 54100, Malaysia; 2Faculty of Applied Sciences, Universiti Teknologi MARA (UiTM) Cawangan Pahang, Jengka 26400, Malaysia; 3Department of Mechanical Engineering, The State University of New York, Korea (SUNY Korea), Incheon 21985, Republic of Korea; 4Department of Mechanical Engineering, Industrial University of Ho Chi Minh (IUH), Ho Chi Minh City 70000, Vietnam; 5Department of Mechanical Engineering, Vocational College, Universitas Gadjah Mada, Yogyakarta 55281, Indonesia

**Keywords:** magnetostriction, magneto-deformation, magnetoactive elastomer, normal force, hysteresis, magnetorheological elastomer

## Abstract

Magnetorheological elastomer (MRE), which is capable of exhibiting magnetostriction in the presence of a magnetic field, has a great potential to be used for the development of sensor devices. Unfortunately, to date, many works focused on studying low modulus of MRE (less than 100 kPa) which can hamper their potential application in sensors due to short lifespan and low durability. Thus, in this work, MRE with storage modulus above 300 kPa is to be developed to enhance magnetostriction magnitude and reaction force (normal force). To achieve this goal, MREs are prepared with various compositions of carbonyl iron particles (CIPs), in particular, MRE with 60, 70 and 80 wt.% of CIP. It is shown that both the magnetostriction percentage and normal force increment are achieved as the concentration of CIPs increases. The highest magnetostriction magnitude of 0.075% is obtained with 80 wt.% of CIP, and this increment is higher than that of moderate stiffness MRE developed in the previous works. Therefore, the midrange range modulus MRE developed in this work can copiously produce the required magnetostriction value and potentially be implemented for the design of forefront sensor technology.

## 1. Introduction

In recent years, there is a constant demand for soft sensors which innately possess desirable features such as high durability and flexibility yet being resilient and robust. The design and development of these soft sensors will therefore warrant searching for appropriate smart materials to overcome some of the limitations of the constituent materials of solid sensors. Among the types of smart materials available to date, magnetorheological elastomer (MRE) has the potential to be implemented in soft sensors due to its ability to adapt to multiscale and dynamic deformations, mechanical compliance and the exceptional magnetostriction phenomenon that is sensitive to the strength of a magnetic field. Magnetostriction phenomenon, or commonly denoted in the literature as magnetostriction [1,2] can be defined as a change in physical dimensions in the presence of a magnetic field. This alteration in dimensions includes a contraction or an elongation, a process within the materials which involves converting the electromagnetic energy into mechanical energy. Both MRE and crystalline ferromagnets display a magnetostriction effect when subjected to a magnetic field. However, the underlying process of such length-changing behavior defers between both materials. In crystalline ferromagnets, the magneto-strain coupling is caused by the quantum spin–orbit interaction [3]. On the other hand, the basis of such effect in MRE is the restructuring or rearrangement of the particles embedded within the matrix to align in the direction of the magnetic field. The degree of elongations and deformations of MRE is largely depending upon the stiffness characteristics of the elastomeric matrix and the magnetic properties of the embedded particles.

Taking into account the variation of probable matrix-particles combinations, previous works on MRE’s composition such as the compliance of the elastomer matrix, volume fraction and distribution of the particles have been conducted in order to fine-tune its magnetostriction response. Among all possible material compositions, a combination of Terfenol-D (TD) particles and Epoxy matrix accounted for the majority of research work on the magnetostriction effect in MRE [4,5,6,7,8]. Such material combination was reported to produce a magnetostriction value up to 0.15% [8]. However, this value was lower than the monolithic form of TD (0.3%) [9]. Other studies have combined polyurethane (PU) elastomers with TD [10] due to better degradation stability than natural rubber and obtained a maximum saturation magnetostriction value of 0.0813% at an applied magnetic field of 0.5 T [11]. Despite the MRE based on TD producing a massive magnetostriction value, some works have combined Galfenol (Fe–Ga) with epoxy and PU matrix [12,13,14] due to lower cost than TD and showed that the value of saturation magnetostriction could reach above 0.03% [15]. Most recently, a combination of iron, cobalt and vanadium (FeCo–2V) was used as the particles to fabricate MRE and obtained magnetostriction value of 0.3731% [16]. In addition to the aforementioned MRE composition, some works have crosslinked micron-sized iron particles with silicone rubbers and produced varying magnetostriction strain. Iron particles are commonly used for MRE and known for their high permeability [17], high saturation magnetization [18] and low remanent [19], while silicone rubber possesses excellent aging properties [20], good thermal and processability [21,22], as well as their low modulus that manages to facilitate the magneto-mechanical interaction [23]. One kind of silicone rubber, which has been every so often adopted as the matrix of MREs in the recent works is polydimethylsiloxane (PDMS). This is due to the excellent adhesive property with iron and low viscosity of its precursor and thus promoting homogeneous particles distribution and ease of MRE fabrication, while also avoiding the possibility of particle agglomeration [24,25,26,27].

Research works pertaining to magnetostriction of MRE based on iron particles and silicone rubber (SRFE) can be found in the literature, and the trend over the year suggests utilizing low-modulus elastomer in order to augment the magnetostriction magnitude, while lowering the magnetic field imposed. As far as particle distribution is concerned, orientation of chain-like structure (anisotropic) produces higher magnetostriction than random isotropic. Bednarek [28] was the earliest pioneer to study MRE based on SRFE more than 20 years ago and reported a relative magnetostriction of about 1% in magnetic fields up to 8 T using silicone rubber with elastic modulus of 0.25 MPa. Afterward, a similar author [29] incorporated pores in the MRE (foam-like) and obtained the maximum magnetostriction of 4.81%. In another study, Guan et al. [1] studied the magnetostriction on isotropic and anisotropic MRE based on SRFE with three different particle volume fractions (15, 20 and 27 vol%) and observed a maximum magnetostriction of 0.0184% for the anisotropic MRE and 0.0134% for the isotropic MRE. From this point onwards, many studies on magnetostriction of MRE seem to employ low stiffness in their samples. This was demonstrated by Abramchuk et al. [30], at which higher magnetostriction magnitude (11.6%) was achieved than previous studies in a MRE based on SRFE with storage modulus of 13.5 kPa. Later on, Diguet et al. [31] also used low storage modulus silicone rubber (47 kPa) to fabricate an isotropic MRE, and were able to produce a magnetostriction magnitude of 9.2%. The trend of using low stiffness MRE was continued by Saveliev et al. [2], in which they obtained the largest measured magnetostriction in MRE based on SRFE (approximately 20%). Such large magnetostriction value is made possible due to the low modulus of MRE. The MRE, which is made of PDMS matrix along with other additives such as plasticizer, has a storage modulus of 30 kPa. They also investigated the effect of the MRE storage modulus on the magnetostriction by fabricating three samples varying in modulus (30, 40 and 50 kPa) and concluded that magnetostriction increased with decreasing modulus of MRE, where the particles’ orientation in a chain-like structure influenced the magnetostriction considerably.

All of the aforementioned works on low-modulus MRE are known to use some form of additives to reduce the stiffness of the matrix. In fact, the fabricated MRE samples have disproportionately lower modulus than the bare silicon rubber. As an example, the low storage modulus of MRE obtained in the work of Saveliev et al. [2] is actually significantly lower than the bare PDMS. Lötters et al. [32] reported that the storage modulus of PDMS fabricated with the standardized ratio of 10:1 with its curing agent was 250 kPa. Such value of storage modulus is expected to increase when magnetic particles are added to the PDMS to fabricate the MRE without using any form of additives such as plasticizer or chain extender modifier. Indeed, it has become apparent that the nature of magnetostriction behavior of MRE is significantly affected based on the constituent materials, particles’ concentrations, distribution and sample shape as well as the elastic modulus of the polymer matrix. Although the use of the low modulus of MRE can produce higher magnetostriction value [2], problems such as short lifespan and low durability should arise when integrating the MRE in some high-performance sensor application [33]. As an example, for a multi-fingered sensor hand of a robot which is required to dexterously grasp and manipulate an object, the use of low modulus of elastomer may cause handling problems and be unable to bear a larger load due to unstable structure, and thus limiting the application to relatively light and soft objects only [34,35]. Additionally, the majority of works on magnetostriction of MRE based on SRFE focuses on the study of a low stiffness elastomer. The popularity of utilizing soft silicon rubber has limited the investigation of MRE with storage modulus higher than 300 kPa. In fact, reported storage modulus values of MRE based on SRFE in the literature range from 13.5 kPa [30] to 200 kPa [1,36] while Young’s modulus takes values from 0.25 [29] to 1100 MPa [7] in magnetostriction studies.

A higher storage modulus indicates that the material can store more energy. As an example, at very low frequencies, the rate of shear is very low, and hence, the capacity to retain the original strength of the media is very high. The rate of shear increases as the frequency and amount of energy input to the polymer matrix chains are increased. Because the storage modulus determines a polymer matrix’s solid-like character, the higher the storage modulus, the more difficult it is to break down the polymer matrix and prolong its durability performance. After removing the applied force, material flow recovery will be greater than a smaller storage modulus value. Taking these points into consideration, the magnetostriction of MRE based on SRFE with storage modulus above 300 kPa is an important matter worthy of investigation. Therefore, this paper presents a set of experiments to study the magnetostriction of MREs with storage modulus above 300 kPa. In order to achieve this target, MRE samples are made utilizing polydimethylsiloxane (PDMS) silicone rubber with different weight fraction of the CIPs, followed by the morphology investigation to show the microstructure characterization. Then, the field-dependent magnetostriction and normal force of MRE samples are evaluated and the results are discussed in the sense sensor applications.

## 2. Materials and Methods

### 2.1. Material and Preparation of MRE

The matrix material used to fabricate the MRE samples was polydimethylsiloxane (PDMS) silicone rubber from SYLGARD™184 (The Dow Chemical Company, Midland, MI, USA). The base matrix consisted of two parts, a curing agent and a low viscosity of liquid silicone rubber (base agent) which smoothen its fabrication process and handling. The density of PMDS-based matrix is 1.1 g/cm^3^. The particles that used the MRE were carbonyl iron particles (CIPs) type OM from BASF (Ludwigshafen, Germany). The CIPs had an average diameter of 5 μm with a spherical shape and were a type of soft magnetic particle. The CIPs’ density was approximately 7.874 g/cm^3^ and the purity of iron in the CIPs was high (99.9%). This type of CIPs was reported to have high saturation magnetization, high permeability and low remanent magnetization [37,38,39]. The experimental procedure for fabricating the PDMS-based MRE is as follows. Firstly, the base agent and hardener of PDMS at a weight ratio 10:1 was firstly mixed evenly using a mechanical stirrer at room temperature for about 10 min. Then, the CIP was added to the mixture and stirred sufficiently for another 10 min. The final mixture was poured into a mold to ensure a 1 mm thickness of samples. The mold then was placed into a vacuum chamber for 30 min to eliminate the unwanted bubbles. The curing process was carried out in an oven at 100 °C for approximately 35 min without the presence of a magnetic field and the mold was left to cool down before removing the MRE sheet from the mold. Afterward, the MRE sheet was cut using a hole puncher into a circular shape with a diameter of 20 mm and thickness of 1 mm for magnetostriction measurements. In this research, three different samples varying in CIPs weight concentrations (60, 70 and 80 wt.%) were prepared. The CIPs concentrations was measured in weight percentages relative to the total weight of the sample. In an effort to meet the selection criteria (storage modulus of MRE above 300 kPa), initial tests were performed on the prepared samples to determine the storage modulus. The range of storage modulus varies from 350 up to 743 kPa depending on the CIP content in the MRE sample. Table 1 summarizes CIP and PDMS compositions of the fabricated MREs and their respective storage modulus.

### 2.2. Microstructure Observation and Magnetostriction Measurements

Field-emission scanning electron microscopy (FESEM) (JEOL JSM 7800F, Tokyo, Japan) was used to observe the microstructure of PDMS-based MREs. For FESEM, the samples were sliced into extremely small pieces (5 × 1 mm) and coated with a thin layer of gold prior to perform the analysis in order to prevent charging of the surface and to provide a homogeneous surface for analysis and imaging. The cross-section of the samples was examined at an acceleration voltage of 2 kV under magnifications up to 1200×. Magnetostriction measurements were conducted using a commercial rheometer (Modal: Physica MCR302, Anton Paar, Graz, Austria equipped with a MR device (MRD 170/1 T). The intensity of the magnetic field was controlled by adjusting the current applied to the electromagnetic coil. The MRE samples were placed between a parallel base plate and the pedestal. The magnetic field was generated perpendicular to the disk plane and going through the samples. In order to ensure a homogeneous magnetic field is being applied on the MRE sample, the two-part metal cover was used to close the magnetic generator (MRD 170 controller). Figure 1a,b show the schematic of the rheometer (MCR302) and the two-part metal cover (MRD 170 controller). It is noted that the H vector arrows representing the flow of magnetic field shown in Figure 1 are an idealized concept of magnetic flow within the rheometer and do not indicate field concentration areas. This kind of ideal working condition to illustrate the magnetic flow within the rheometer is normally provided by the rheometer manufacturing company (Anton Paar in this work). In addition, it is remarked that the research on the simulation of actual field flow is beyond the scope of this work.

The magnetostriction measurement was carried out as follows. The current (I) was set using the computer and supplied to the electromagnet coil to generate a uniform magnetic field, H on the MRE. The current between 0 to 3 A was varied to achieve magnetic field excitations in the range of 0 to 0.6 T during the tests. Table 2 describes the current magnitude relative to the magnetic field. The change in length (∆l) of the sample (magnetostriction) was detected by the parallel base plate and determined from the difference between the initial (l) and the final length of the sample as given in Figure 1c. In comparison with the most commonly used strain gauge method, magnetostriction measurements via rheometer method are tailored toward samples having extremely thin thickness (approximately 1 mm) and being small in diameter. In such cases, attachment of a conventional strain gauge to a soft and small MRE can possibly alter its deformation and magnetostriction measurement. Conversely, the existing strain gauge method may work well for classical magnetostrictive materials which are usually relatively rigid (the elastic modulus is in range of 10^11^ Pa).

## 3. Results and Discussion

### 3.1. Morphology Characteristic

Figure 2 shows the cross-section surface imaging for all three microstructures of the MRE samples. A random distribution of CIPs which forms an isotropic structure was observed homogenously distributed and well embedded within the elastomer matrix. However, some voids seemed to appear as well. Figure 2a represents the MRE with 60 wt.% with a little agglomeration of CIPs, though the level of agglomeration as well as voids increase with higher CIP concentration. The same observation is noticeable in Figure 2b,c for samples with 70 and 80 wt.% CIPs, respectively. Furthermore, the higher concentration of CIPs in MRE, the denser the matrix was occupied with CIPs, indicative of lesser space and closer distance between CIPs.

### 3.2. Effects of Particles’ Weight Percentage on Magnetostriction of MRE

In general, a particular feature of the material employed in sensors is the ability to produce a reaction force in order to interact with the external stimulus and to generate movement. In this study, when the MRE sample was under the presence of magnetic field, the CIPs embedded within the matrix rubber become magnetized. Concurrently, the magnetic moments of the CIPs tended to rearrange themselves to be realigned in the direction of the applied magnetic field. This resulted in interaction between the CIPs and the forming of a chainlike structure, which consequently led to deformation or magnetostriction of MRE. An induced/reaction force, which is also called a normal force, was generated at the same time. Thus, to better understand the reaction force of the material, the normal force was studied alongside the magnetostriction under various strengths of magnetic field (from 0 to 0.6 T). The graphs of magnetostriction and normal force as functions of magnetic fields are presented in Figure 3.

The curves of magnetostriction as shown in Figure 3a show that all MRE samples are experiencing an increase in magnetostriction under the influence of a magnetic field, indicative of the fact that the samples were elongated. The general behavior of such an increase in magnetostriction can be divided into three main stages. In the initial state, all MRE samples have not exhibited magnetostriction when a magnetic field was first imposed until up to approximately 0.1 T. A similar occurrence has been reported in the past work [2] and such benign effects may be attributed to the higher constraining force of the elastomer matrix than the magnetized CIP in MRE, which prevents magnetostriction from occurring at a low magnetic field. From 0.1 until 0.3 T, there was a slight increase in magnetostriction for all MRE samples, and at this range of magnetic field, the magnetostriction curves almost coincided, which was the indication of similar increasing rate. A magnetic field of 0.3 T marked the onset of diverging curves of magnetostriction, where after that point, MRE 80 wt.% displayed a steeper increase in magnetostriction than MRE 70 and 60 wt.%. The magnetostriction curves culminated at a magnetic field of 0.6 T with MRE 80 wt.% exhibiting the highest magnetostriction value (0.075%), followed by the MRE 70 wt.% (0.05%) and MRE 60 wt.% (0.04%). By comparison, the magnetostriction value obtained in this study has shown an improvement of 4.5-fold compared to the previous study [1] for similar CIPs structure (isotropic) and 3-fold for the aligned CIPs (anisotropic), while applying 25% less maximum magnetic field.

On the other hand, Figure 3b shows the change of normal force as a function of a magnetic field. Overall, all MRE samples with various CIPs weight concertation demonstrated a gradual increase in the normal force as the magnetic field was increased. Such an increasing trend was interestingly analogous to the increase in magnetostriction value as increasing the magnetic field. The rate of increase, as well as the curves’ steepness of the normal force, were qualitatively similar to the magnetostriction curves. The MRE with 80 wt.% of CIPs generated the highest maximum normal force, which was larger than 1.5 N at 0.6 T. Such force was approximately 40 and 70% higher than the MRE sample with 70 and 60 wt.%, respectively. In terms of sensitivity of the MRE material, the sample with 60 wt.% CIP demonstrated a high sensitivity towards the normal force, which was capable of detecting as low as 0.01 N at 0.072 T and thereby making it suitable for sensor applications, since high sensitivity is required during usage. Table 3 summarizes the relative magnetic field required of all samples to detect a normal force of 0.01 N.

The different rates of increase in magnetostriction and normal force at different intensities of magnetic field could be attributed to the critical value of the magnetic field strength [40,41]. At the initial state, the elastic interactions of the matrix were more dominant than the magnetic interactions (between the magnetized CIPs) due to the magnetic field strength being lower than the critical value. This led to restrictive movement of CIPs. This phenomenon has been observed in [40] and was explained by the fact that the magnetically-induced CIPs were insufficient to overcome the elastic energy barrier of the elastomer matrix in order to elongate. At a certain critical value of the magnetic field strength, the elastic interactions were comparable with magnetic interactions. As the magnetic field exceeded a particular threshold of magnetic field (higher than the critical value), the magnetic interaction was prevailing over the elastic interactions, which accelerated the movement of CIPs for realignment process. As a result, the rate of increase in magnetostriction and normal force became faster. Figure 4 presents a comparison of the magnetostriction magnitude with other previous studies regarding the storage and Young’s modulus of MRE. To some extent, the difference in the constituent materials, particles’ concentrations, distribution and the imposed magnetic field on the MRE made fair and direct comparison between results difficult. However, a superficial analysis of the orders of magnitude of the magnetostriction and the respective modulus of the MRE revealed appealing results. As shown in Figure 3, the magnetostriction values in the present work are nearly 0.08% with storage modulus of 743 kPa, which well eclipses the previous results. Thus, it indicated that the aspect ratio of a sample could affect the magnetostriction of MRE to a certain degree. Additionally, the range of storage modulus from 300 to 750 kPa was an ideal range for potential applications in force sensors [42,43] and actuators [44,45,46], since moderate stiffness could ensure high durability of the sensor device, while maintaining sufficient magnetostriction magnitude.

It is remarked here that the distribution of the CIP would affect the magnetostriction. A couple of the previous works have reported that pre-aligning the CIPs along the direction of magnetic field would increase the magnetostriction value, whereas pre-aligning the CIPs in the transverse direction would decrease the magnetostriction value. However, this phenomenon has not always been the case, since other works demonstrated lower magnetostriction value compared to random distribution. This is attributed to the increase in MRE stiffness, which inhibits elongation.

### 3.3. Normal Force under Cyclic Magnetic Loading at Different Strains

The following sets of experiments were designed from the viewpoint of practical sensor applications. Such experiments could be used to evaluate the reusability and reproducibility of the generated normal force of the MRE. In order to simulate the material used under real settings, the MRE was subjected to continuous cyclic increase and decrease in the magnetic field between the maximum field amplitude (*H* = 0.6 T) and zero magnitude. The stepwise increase and decrease in the magnetic field were consecutively repeated several times (10 cycles of stepwise increase and decrease) at shear strain of 0.01%, 0.1% and 1%. In addition, previous works have reported that hysteresis phenomena such as storage and loss modulus and magnetostriction were evident in MRE [40,41,47,48]. Therefore, it should be expected that there would be hysteresis in the normal force as well. In this section, the MRE of 80 wt% was chosen for a further hysteresis analysis since it showed the highest normal force value. Figure 5 shows the change in normal force of MRE with 80 wt.% CIP as function of stepwise magnetic fields between zero field and 0.6 T. The color transition from light to dark blue of curves denoted the stages of the cyclic experiment, where the light and the dark colors indicated the initial and final stage of the magnetic cycle, respectively. In Figure 5a, the relative curve of increase and decrease in normal force as a result of the imposing and removing of a magnetic field is evidently nonlinear. The path corresponding to the ascending and descending of normal force as a function of magnetic field at strain amplitude of 0.01% differs, indicative of hysteresis behavior in the MRE material. It is observed that the descending branch of normal force was larger than the ascending branch. Qualitative comparison between the initial and each subsequent cycle of magnetic field loading showed the curves shifted upward and tended to reach saturation in the last few cycles. At strain amplitude of 0.1%, the change in normal force as a function of magnetic load has similar characteristics to the strain of 0.01% as given in Figure 5b. It is seen that the normal force at strain amplitude of 0.1% increases with the increasing magnetic field intensity and decreases with decreasing magnetic field intensity. Overall, all curves’ shapes of ascending and descending of normal force were reproduced very well for the three-shear strain imposed. A quantitative comparison between every cycle of magnetic field showed little variations in patterns of ascending and descending of normal force. The maximum normal force, however, decreases with increment of shear strain, and this was due to the shear stress on the MRE material. With increasing shear strain, the inter-CIPs distance decreases and under the influence of a magnetic field, the magnetic moment in every CIP was induced and rotated. As a result, CIPs tended to align in the direction of the external magnetic field and induced an attraction force within each CIP. The closer distance between CIPs led to a greater force of attraction between the CIPs. However, at a larger shear strain, the structure formed between the magnetized CIPs might break and consequently reduce the elastic energy boundary. Previous works [49,50] have reported that the elastic modulus of MRE decreases with increasing shear strain, and this caused the normal force to decrease as well.

Similar phenomenon such as the descending branch of normal force above the ascending branch could also be seen, as well as the upward shift of normal force curves as the number of loading cycles increases, though the maximum normal force value decreases to nearly 0.4 N when imposing higher shear strain (0.1%) than previous shear strain amplitude (0.01%). In addition, increasing the shear strain amplitude to 1% hardly affected the curve of normal force as given in Figure 5c. Regardless of whether the comparative basis was the nonlinear shape of the ascending and descending of normal force or the pronounced upward shift of normal force curves with the increase in loading cycles, the MRE material nonetheless exhibited hysteresis phenomenon at a strain amplitude of 1%. The increase in strain amplitude from 0.1% to 1% resulted in lower magnitude of maximum normal force (slightly above 0.3 N at 1% strain), and while this figure was considered the lowest maximum normal force among all shear strain amplitudes, the highest maximum normal force was evidently produced at strain amplitude of 0.01% (slightly above 0.5 N). In order to quantify the hysteresis phenomenon in MRE, the employment of the changes of the maximum normal force (λ) during the course of increasing/decreasing in the magnetic cycles was examined. In particular, the shift, λ, is calculated by subtracting the relative maximum normal force in the nth (F_n_) from the maximum normal force (F_1_) in the initial cycle. The total subtraction value (∆F) is then divided by F_1_. The constitutive equation to determine λ is given by:λ = (F_n_ − F_1_)/F_1_,(1)

Figure 6 shows the curves of parameter λ, which commendably reflects the changes of the maximum normal force of MRE in the magnetic cycles for the three different strain amplitudes imposed. For all three data sets, the parameter λ tended to grow with the continuous increase in number of cycles until a saturation point was reached. At shear strain of 0.01%, the increases of parameter λ were measured on the order of 18% and saturated at the 7th cycle. Meanwhile, under the shear strain of 0.1%, the curves of parameter λ demonstrated an almost similar growth trend to the strain of 0.01%, though the former marginally instigated a larger shift of normal force magnitude (about 5% higher).

Significant changes could be seen within the first cycle and only moderate changes (3%) for the following cycles. A linear increase in parameter λ could also be observed with the corresponding rates of its growth per cycle, which was approximately 3% from the second cycle up to the saturation point (7th cycle). The hysteresis nature of the MRE was more prevalent under shear strain of 1%, with parameter λ measuring a larger value than its counterpart (about 8% higher than 0.1% strain and 14% higher than 0.01% strain). Furthermore, at 1% shear strain, the parameter λ continued to grow after the 7th cycle and saturated much later at the 9th cycle. The high magnitude of λ indicated that the displacement of interfacial slipping between the CIP and the matrix was large enough to warrant shifting of maximum normal force, and that the relative shift of maximum normal force accumulated with the increases of cyclic loading. A comparison between the first cycle and second cycle has a significant increase in parameter λ for all values of shear strain and such behavior was more predominant with higher shear strain. A considerable change of λ in between the first cycle and second cycle were congruent with the initial (increasing/decreasing magnetic field) curve of normal force from the subsequent cycles as depicted in Figure 6. Similar phenomena have been reported for magnetic hysteresis behavior [51], dynamic modulus [40], normal force [40,48] and dielectric permittivity [52]. A large change of λ between the first and second cycle implied that principal restructuring of the CIPs occurred during the first cycle, which resulted in major shift of maximum normal force, but minor in the subsequent cycle. The underlying reason for the hysteretic behavior in MRE could be attributed to the dependence of CIPs restructuring/restructuring on the history of magnetic loading-unloading.

An examination on the CIP’s state of motion might reveal what likely transpires on the microscale level and possibly explain the working mechanism behind the MRE hysteresis phenomenon. Theoretically, the MRE hysteresis phenomenon was related to the transition of phases of the continuous loading cycle. At zero magnetic field, the CIP and matrix of the MRE were in equilibrium, and when exposed to a magnetic field, the CIP tended to move along the direction of the magnetic field from its initial position, while being pushed by the attraction from the nearest neighboring CIP. The distance between the CIP and its origin position grew, while the distance between CIPs closes and the CIPs interacted. As a result, the elastomer molecular chain stretched longer, producing a magnetostriction effect in the MRE sample. Correspondingly, as the magnetic field was gradually decreased to zero from its maximum value with a constant shear, the CIP was inclined to return back to its initial position driven by the elastic force of the elastomer matrix. The CIP, however, has not immediately returned to its initial position, but relocated to a new position, which led to some remanent appearing. This caused the initial value of normal force to shift as compared to the previous cycle. Over the course of continuous cyclic ascending and descending of the magnetic field, the remanent along with the stretching of the elastomer molecular chain grew, thus shifting the maximum normal force to a higher value until a state of saturation was achieved. Additionally, larger shear strain caused larger bond rupture between the CIP and the polymer matrix, which was an interesting corollary to the increase in distance between the CIP and its initial positions. Consequently, the displacement of interfacial slipping between the CIP and the matrix was also increased with larger shear strain, and hence, a larger shift of maximum normal force taken place.

Table 4 summarized the field-dependent magnetostriction of MRE currently available in the literature along with the storage modulus value. A direct and shallow comparison between the studies demonstrates a disproportionately higher magnetostriction magnitude for the low modulus of MRE than the midrange modulus of MRE. In spite of this, an assessment on other factors such as the costs of the manufacturing of such materials, the durability performance and the types of sensor application may reveal an interesting outcome.

Figure 7 is a representation of the state-of-the-art robotic gripper utilizing MREs as a force sensor. The robotic gripper, which has a simple structure, can grasp diverse arbitrary objects, while the force sensor attached to the end effector of the gripper can measure the reaction forces. When assigned to repetitive tasks and prolonged use of the robotic gripper, several issues which are often associated with the durability of MRE such as elastic stretching, cross-link disengagement, inelastic deformation, structural shift by phase transformation, microplasticity and microphase separation can arise [53,54]. In particular, the implementation of low-modulus MRE on the robotic gripper can theoretically accelerate these processes within a finite interval of time, and consequently, permanent deformation or a non-reversible change in shape of the MRE may ensue. Such phenomena can be further explained by the fundamental principle of storing energy performance in MRE. A material with a higher storage modulus has the potency to store more energy. For example, the rate of shear is low at very low frequencies, hence, the capability to retain the original strength of the media is high. As the rate of shear increases with frequency, the amount of energy stored to the polymer matrix chains also increases. The storage modulus, which primarily determines the elastomer matrix’s solid-like character, also plays an important role in the strength and rigidity of the material. A midrange modulus MRE possesses better strength and is harder to break down its elastomer matrix than the low-modulus MRE. Shape-retaining and recovering after removing the applied force will be better with a midrange modulus MRE than that of a low-storage modulus. Therefore, the implementation of a midrange modulus MRE as a sensor can improve its durability performance and prolong its life cycle.

## 4. Conclusions

The current work presented comprehensive investigation of magnetostriction in MRE with moderate modulus by imposing the homogeneous magnetic field. Apart from magnetostriction, the MRE has the capability to produce a reaction force called the normal force. The MRE samples with different concentrations of CIPs ranging from 60 to 80 wt%. were successfully prepared. The highest magnetostriction achieved by the MRE of 80 wt% CIP was 0.075% and the normal force produced was 1.56 N at 0.6 T. The measured magnetostriction was considered as an improvement over the MRE with similar range of modulus found in the literature. Nevertheless, the magnetostriction improvement is regarded as not outstanding (approximately 3-fold higher than the previous work [1] for the aligned CIPs and 4.5-fold for similar CIPs structure). This would certainly depend upon the comparison of the relevant fabrication and experimental parameters such as the intensity of magnetic field (25% less maximum magnetic field than [1]), stiffer MRE samples (nearly three-fold higher storage modulus than [1]) and the structure of the embedded CIPs in MRE. As such, the trivial enhancement of magnetostriction achieved can still be particularly valuable in soft sensor design that requires a corresponding increase in the magnitude. Meanwhile, the results demonstrated that the MRE with 60, 70 and 80 wt% showed a trend of increase in the magnetostriction percentage and normal force that was parallel with the increased concentration of CIPs. With regard to the sensitivity of the MRE material, as low as 0.01 N of normal force can be detected while employing a small amount of magnetic field (0.072 T). In general, high sensitivity with change in magnetic field is sought after in soft sensor applications. In particular, a soft robotic gripper with an MRE will be able to have an infinitesimal deformation upon being triggered by a magnetic field and thus easier to predict the response of closing patterns accurately upon contact or touch with the object. Additionally, such a trait is exceptionally useful for selecting grasping patterns according to the object nature.

On the other hand, the hysteresis analysis showed that the maximum normal force produced by the MRE with 80 wt% CIP increases with the number of cycle loads of magnetic field until a saturation point was reached. The change of maximum normal force was dependent on the shear strain amplitude. In other words, higher amplitude of shear strain resulted in greater change of maximum normal force. Based on the presented results, the midrange modulus MRE herein could be considered as a viable candidate for specific sensor applications, namely, force sensors [42,43], since moderate stiffness could ensure high durability of the sensor device, while maintaining sufficient magnetostriction magnitude. Characterization of various temperature effects on the magnetostriction performance will be a beneficial starting point for future work given that the utilization of MRE in sensor will not be restricted to a particular scenario, yet able to function even in extreme conditions. It is finally noted that manufacturing of a specific MRE-based sensor device and its experimental performances such as accuracy and signal producing remain as future works.

## Figures and Tables

**Figure 1 micromachines-14-00767-f001:**
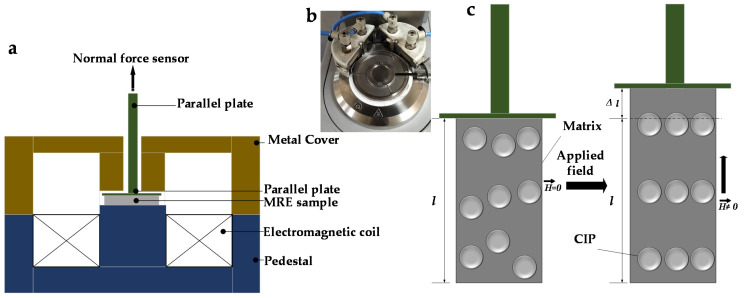
Experimental set-up of magnetostriction measurement (**a**) Schematic illustration of rheometer, (**b**) the two-part metal cover (MRD 170 controller) and (**c**) schematic illustration of the MRE under a magnetic field: *H* is the magnetic field intensity, *l* is the length of sample.

**Figure 2 micromachines-14-00767-f002:**
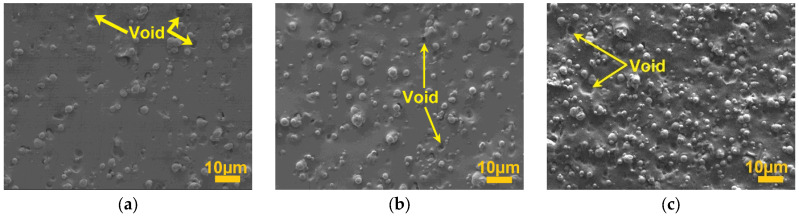
Microstructure of PDMS MREs under magnifications of 1200× with CIPs of (**a**) 60, (**b**) 70, and (**c**) 80 wt.%.

**Figure 3 micromachines-14-00767-f003:**
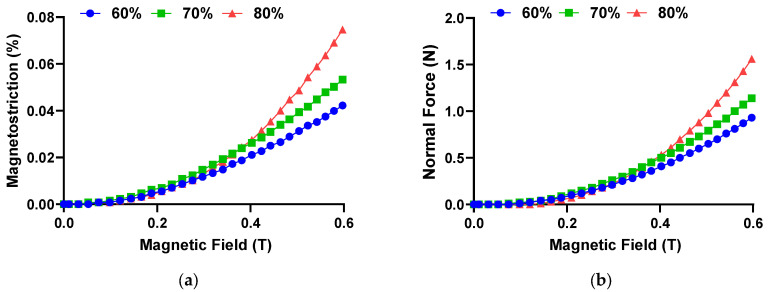
The curves of (**a**) magnetostriction and (**b**) normal force versus the magnetic field with various concentrations of CIPs.

**Figure 4 micromachines-14-00767-f004:**
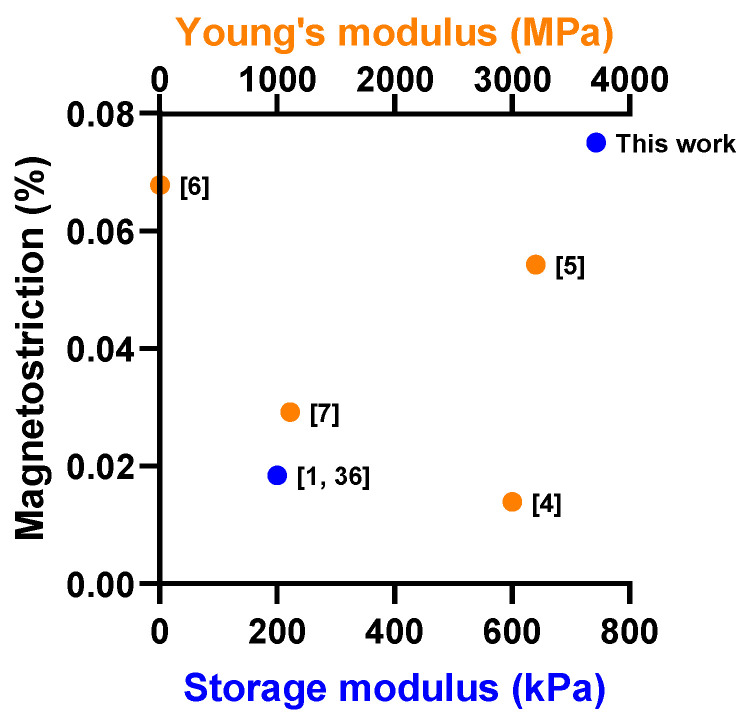
Comparison of magnetostriction obtained versus the storage/Young’s modulus [1,4,5,6,7,36].

**Figure 5 micromachines-14-00767-f005:**
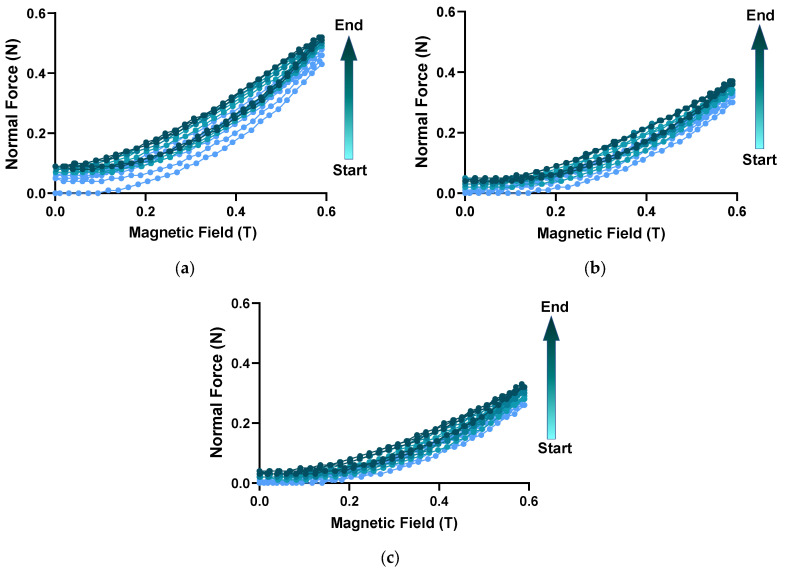
Comparison of normal force at shear strain of (**a**) 0.01%, (**b**) 0.1% and (**c**) 1% versus the magnetic field.

**Figure 6 micromachines-14-00767-f006:**
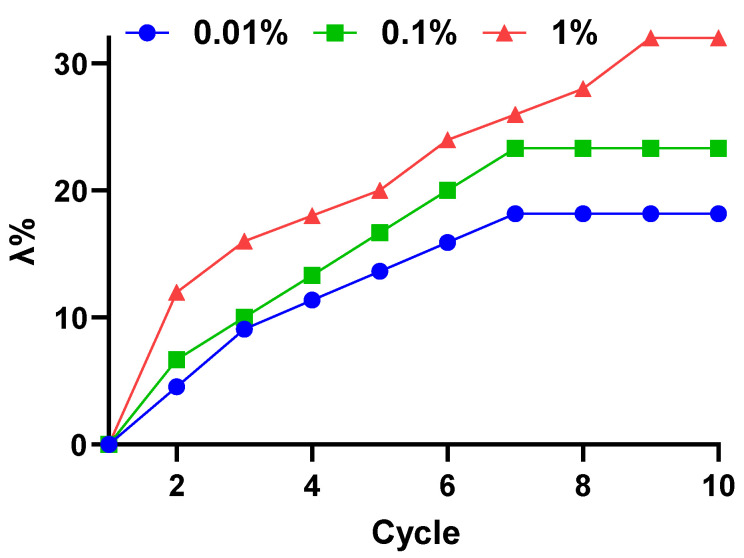
Dependence of the parameter λ on the number of cycles loading.

**Figure 7 micromachines-14-00767-f007:**
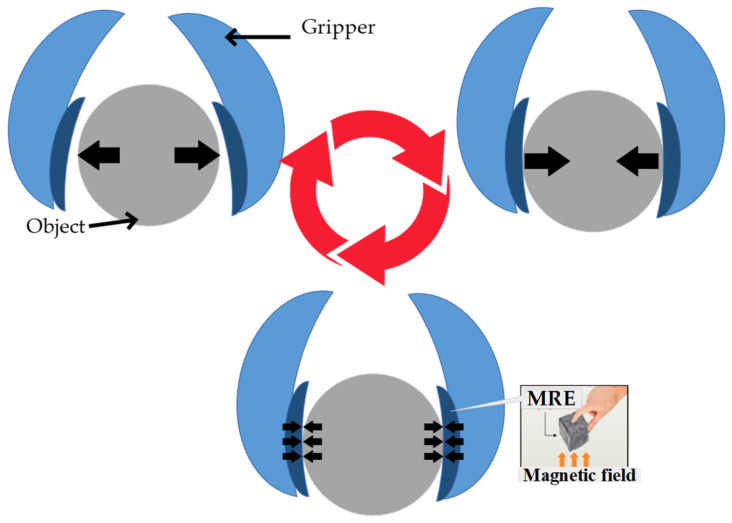
The cycle of use of soft robotic gripper with MREs.

**Table 1 micromachines-14-00767-t001:** Details on the proportions of components and storage modulus of MRE samples.

Samples	Particle’s Weight (%)	Sylgard 84 (10:1 wt%)	Storage Modulus (kPa)
MRE 60 wt.%	60	40	350
MRE 70 wt.%	70	30	680
MRE 80 wt.%	80	20	743

**Table 2 micromachines-14-00767-t002:** The current magnitude relative to the magnetic field.

Current (A)	Magnetic Field (T)
0	0
1	0.21
2	0.39
3	0.61

**Table 3 micromachines-14-00767-t003:** Summary of the sensitivity of the MRE material to detect 0.01 N.

Samples	Magnetic Field (T)
MRE 60 wt.%	0.72
MRE 70 wt.%	0.74
MRE 80 wt.%	0.78

**Table 4 micromachines-14-00767-t004:** Selected studies investigating the magnetostriction of MRE along with the storage modulus value.

Modulus (kPa)	Magnetostriction	Reference
13.5	11.6%	Abramchuk [30]
47	9.2%	Diguet [31]
47	20%	Saveliev [2]
47	0.0134%	Guan [1]
350–743	0.04–0.075%	This work

## Data Availability

The raw/processed data required to reproduce these findings cannot be shared at this time as the data also form part of an ongoing study. In future, however, the raw data required to reproduce these findings will be available from the corresponding authors.

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
