# Peer review of "Magnetostriction Enhancement in Midrange Modulus Magnetorheological Elastomers for Sensor Applications"

_micromachines, 2023, doi:10.3390/mi14040767_

Round 1
Reviewer 1 Report
The MRE with a storage modulus above 300kPa has been analyzed in the manuscript, which shows the potential applications in the sensing area.
To make the manuscript understandable, there are some suggestions to the authors:
1. If the distribution of the CIP would affect the experiment results?
2. The accuracy of the proposed module for the sensing is not clear, please add such statement in the manuscript.
3. Compared with the Guan[1] method, the improvement is not outstanding, please address this issue in the revised manuscript in detail.
Reviewer 2 Report
​The authors have studied in details the Magnetostriction in Midrange Modulus Magnetorheological Elastomers. The findings are interesting and scientifically proven. I recommend it for publication after addressing following points.
1. Please check the unit for density should be properly written e.g. line 155.
2. Authos should desciribe the magnetostrction measurement process in detail and exaplin how it is differnent or advantageous than the most commonly used strain gauge method.
Thanks and Regards
Round 2
Reviewer 1 Report
The response to the comment is sufficient enough for the reviewer.
Author Response
In fact, there is no more comment to be reflected as clearly written at the end of this sheet: The response to the comment is sufficient enough for the reviewer